

# Effects of harvest height and time on hay yield and quality of some sweet sorghum and sorghum Sudangrass hybrid varieties

Fırat Alatürk

Department of Field Crops/Faculty of Agriculture, Çanakkale Onsekiz Mart University, Canakkale, Türkiye

## ABSTRACT

**Background:** This experiment was conducted in the Research and Application Field of Canakkale Onsekiz Mart University, Faculty of Agriculture, during the 2020 and 2021 summer period. The objective of this experiment was to determine the effects of different harvesting heights on forage yields and crude ash, fat, protein, and carbon and nitrogen content of leaves and stalks of sweet sorghum (SS) and sorghum sudangrass hybrid (SSH) cultivars.

**Methods:** Nutri Honey and Nutrima varieties of SSH and the M81-E and Topper-76 varieties of SS were used in this study. The experiment was conducted using the randomized complete block design with four replications. The main plots each included two early and late varieties of SS and SSH cultivars, while the subplots were used to test different harvesting heights (30, 60, 90, 120, 150 cm) and physiological parameters of each crop.

**Results:** The results of this study showed that dry forage yields increased with plant growth, with the amount of forage produced at the end of the growth cycle increasing 172.2% compared to the early growth stages. Carbon (C) content of leaves decreased by 6.5%, nitrogen (N) by 46%, crude protein (CP) by 54%, crude fat (CF) by 34%, while crude ash (CA) content increased by 6% due to the increase in plant height harvest. At the same time, in parallel with the increase in plant height at harvest, the nitrogen content of the stems of the plants decreased by 87%, crude protein by 65%, crude ash by 33% and crude fat by 41%, while the carbon content increased by 4%. As plant height at harvest increased, hay yield increased but nutrient contents of the hay decreased. However, the Nutrima, Nutri Honey and M81-E sorghum cultivars, harvested three times at heights of 90 to 120 cm, are recommended for the highest yield.

# INTRODUCTION

The livestock sector has rapidly changed in Türkiye. Cattle breeding has increased to meet demand for meat, milk and dairy products. More cattle necessitate the cultivation of more forage crops in larger areas at lower costs without competing with food production. Sorghum sudangrass hybrid (SSH) cultivars play an important role in animal nutrition because of their ability to rapidly regrow after harvesting. Lignification is the main factor

Corresponding author
Fırat Alatürk, alaturkf@comu.edu.tr

limiting the digestibility of plant cell wall polysaccharides *in vitro* (*Jung, Samac & Sarath, 2012*) and also its *in vitro* digestibility (*Reddy et al., 2005*). Fresh and dry forage of sorghum varieties are used as silage and haylage as well as for grazing purposes (*Undersander, 2003*; *Avcıoğlu et al., 2009*). Sorghum has been characterized as the "camel of the plant kingdom" because of its high tolerance to temporary drought conditions and its ability to regrow after drought conditions have disappeared (*Sanderson et al., 1992*; *Açıkgöz, 2001*). Sweet sorghum's drought tolerance and high-water use efficiency make it a good option for global warming and drought scenarios and an important forage crop for silage production and energy agriculture in Türkiye (*Yücel et al., 2017*). Since sweet sorghum has a high ethanol yield (*Bayram & Turgut, 2015*), research about this crop has generally focused on its potential in sugar and ethanol production. To maximize efficiency, it is important to determine the optimal harvesting times of SSH varieties in both fresh and dry forage forms. These varieties can produce 1–1.5 tons of fresh forage/ha with 4–5 cuttings in summer, main crop growing conditions and 2–3 cuttings in second crop growing conditions (*İptaş, Brohi & Aktaş, 2001*; *Salman & Budak, 2015*). The reason for investigating different harvesting heights in SS and SSH cultivars is that the harvesting height in plants provides ideas about the development period and its nutritional value of the plant and the amount of the hay it produces. Harvesting time directly affects the quality of hay as well as the regrowth and ultimate yield power. The lower, the plant height at the time of harvesting, the higher the quality of the hay, but the lower or lack of sufficient storage material for regrowth, the slower the regrowth and hence the lower the yield in the second and subsequent harvesting (*Ansa & Garjila, 2019*). In SSH, there is an increase in the number of tillers produced by the plant due to the increase in the number of harvestings, but the dry matter yield decreases up to 50% due to the decrease in photosynthetic tissues (*Sowiński & Szydełko, 2011*). The number of harvestings also affects the yield and quality of the hay produced by the plant (*Rahman, Fukai & Blamey, 2003*). Similarly, there are studies carried out by some researchers showing that dry matter yield decreases with the increase in the number of harvesting (*Lee, 2005*; *Uher et al., 2005*). While high quality hay is obtained when harvesting is done in the early periods, as the harvest time is delayed, the hay becomes coarser and its nutritional value decreases (*Lang, 2001*). Studies on optimal harvesting times, especially those specific to grazing management, are extremely limited. This study was conducted to determine the best harvesting and grazing practices of SS and SSH cultivars for all use cases.

## MATERIALS AND METHODS

### Study location

This study was carried out in the Research and Application Field of Canakkale Onsekiz Mart University, Faculty of Agriculture during the 2020 and 2021 growing seasons. The long-term average temperature of Canakkale province is 15.09 °C according to the Turkish General Directorate of Meteorology. Average temperatures on research years were recorded as 17.01 °C in 2020 and 17.58 °C in 2021, which were both above average. Average precipitation in the area from the first week of May to the last week of October is 149.9 mm. Total precipitation was higher than the average for both years of the study, with

157.5 mm of precipitation measured in 2020 and 201.2 mm measured in 2021 in this time period (Fig. 1).

## Soil sampling

Soil samples were taken at a depth of 0 to 30 cm from many spots throughout the study area and then combined to create a representative sample of the area. Soil samples were analyzed in the soil testing laboratory of the Canakkale Onsekiz Mart University, Science and Technology Application and Research Center (ÇOBİLTUM) according to the methods outlined by *Müftüoğlu, Türkmen & Çıkılı (2012)*. Soils in the research area had a clay-loam texture and were neutral in terms of soil reaction. The soils were determined to be medium calcareous, medium in organic matter, medium in phosphorus content, and deficient in potassium (Table 1).

## Experimental design, treatments details and crop management

Seed sowing was carried out on 16th May in the first year (2020) of the study and on 5th May in the second year (2021). Annealing irrigation was performed before sowing and then the area was plowed deeply with a plow. The seed bed was prepared by pulling a cultivator and disc harrow, and then 1 kg of nitrogen, phosphorus and potassium per hectare were added to the soil with composite fertilizer (15-15-15) before deep plowing. Ammonium sulfate was applied as surface fertilizer at a rate of 50 kg nitrogen per hectare immediately after seed sprouting (*Avcıoğlu et al., 2009*). Soil samples were taken from the experiment plots and then analyzed before fertilizer application. Plants were then irrigated with drip irrigation and irrigation frequency was adjusted based on air temperature and precipitation. In July and August, plants were irrigated about every 7 days. Weeds that emerged during the research period were removed manually by individually removing weeds growing in the rows and hoeing weeds between the rows. This experiment used a randomized complete block design with four replications. The main experiment plots included the cultivars and the subplots were used to test different harvesting heights. Sweet sorghum (SS) cultivars were sown 8 cm apart in rows with 70 cm between rows. Sorghum sudangrass hybrid (SSH) cultivars were also sown 8 cm apart, but with 35 cm between rows (*Mahmood & Honermeier, 2012*). Experiment plots were 5 m in length, with SS cultivar plots arranged in four rows and SSH cultivar plots arranged in six rows. There was no space dividing the experiment plots, but the experiment blocks were divided by a distance of 1 m.

## Hay parameter

Two cultivars of sweet sorghum (Topper-76 and M81-E) and two cultivars of hybrid sorghum sudangrass (Nutrima and Nutri Honey) were used as materials in this study, as shown in Table 2. The cultivars of sweet sorghum used in this study were developed at the University of Nebraska and are among the cultivars considered to be promising based on previous research conducted in Türkiye (*Yücel et al., 2017*). The SSH cultivars used in this study are the registered cultivars sown in Türkiye.
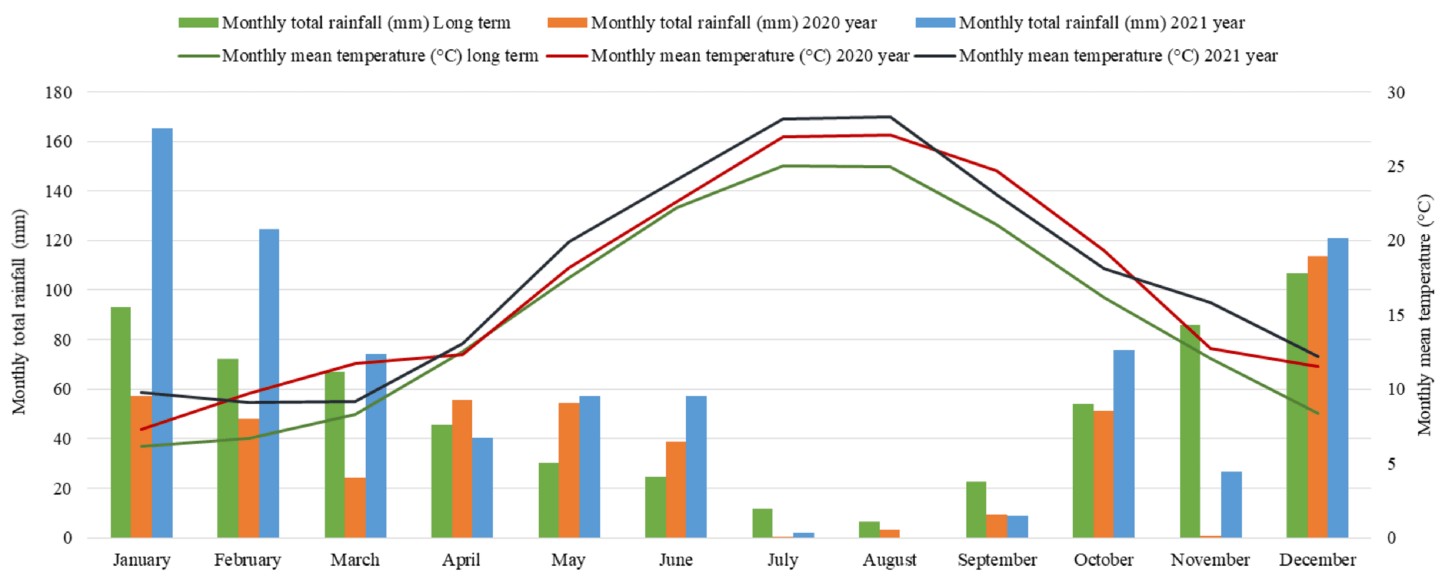

**Figure 1  Meteorologic data for the Canakkale province including long-term averages and data from the study years.**

**Table 1  Soil properties of the research area.**

|  | Soil properties (%) | pH | E.C. (mS/cm) | Lime (%) | Soil organic carbon content (%) | P (mg/kg) | K (mg/kg) |
|---|---|---|---|---|---|---|---|
| Sample 1 | 70 | 7.50 | 0.85 | 8.65 | 1.89 | 66.08 | 358.8 |
|  | Clay-loamy | Slightly alkaline | Saltless | Medium chalky | Poor | Poor | Poor |
| Sample 2 | 65 | 7.35 | 0.88 | 7.69 | 1.95 | 54.88 | 337.9 |
|  | Clay-loamy | Neutral | Saltless | Medium chalky | Poor | Poor | Poor |
| Sample 3 | 68 | 7.31 | 0.95 | 9.16 | 1.78 | 69.44 | 385.5 |
|  | Clay-loamy | Neutral | Saltless | Medium chalky | Poor | Poor | Poor |
| Mean | 67.7 | 7.39 | 89.3 | 8.50 | 1.87 | 63.39 | 360.7 |
|  | Clay-loamy | Neutral | Saltless | Medium chalky | Poor | Poor | Poor |

**Table 2  Sorghum cultivars used in this study and their characteristics.**

| Species | Cultivars | Organization name | Purpose of production | Maturity |
|---|---|---|---|---|
| Sweet sorghum | Topper-76 | Nebraska University | Syrup, ethanol | Mid late |
| Sweet sorghum | M81-E | Nebraska University | Syrup, ethanol, silage | Late |
| Sorghum × sudangrass | Nutri Honey | Alfa Seed | Forage and grazing | Mid early |
| Sorghum × sudangrass | Nutrima | Royal AgriLife | Fresh forage, silage, grazing | Late |

Hay yields were obtained by first removing 50 cm sections from the beginning of the plots that were considered edge effects during plant sampling. The side rows were also included in the harvesting area since there was no space between the plots. When the plants reached the planned harvesting heights, the 5.6 m² harvest area was mowed with a harvester machine and/or sickle, leaving 15 cm of stubble, and the harvested plants were

weighed immediately with a hand scale to obtain their fresh weight (*Lang, 2001*). The obtained values were calculated as fresh hay yield/plot in kg/da. Samples weighing over 1 kg were then taken from these fresh plants, placed into paper bags and brought to the laboratory. The samples were separated into stems, leaves and clusters in the laboratory and air dried. They were then moved to a drying oven set at 60 °C until sample weight was constant (for 48 h) and then the samples were weighed (*AOAC, 2000*). Dry matter ratios (%) were determined by proportioning the weights of the dried samples to their fresh sample weights, as follows:

$$\text{Dry matter ratio}\,(\%) = (\text{Dry hay yield}/\text{Fresh hay yield}) \times 100 \tag{1}$$

whereas, the dry hay yields were obtained first and then multiplied with fresh hay yield and dry matter ratio, and then calculated as kg/da:

$$\text{Dry hay yield}\,(\text{kg/da}) = \text{Fresh hay yield}\,(\text{kg/da}) \times \text{Dry matter ratio}\,(\%) \tag{2}$$

The hay yield of the rangeland in kg/ha was calculated by weighing the hay, and the average was taken. Plant samples were harvested from the rangeland and ground. Then, the crude protein ratios of the samples were determined using the Kjeldahl method (*Bremner, 1960*). C and N ratios of the plant samples were determined by the Eastern Anatolia High Technology Application and Research Center (DAYTAM). A LECO brand CHNS-932 analyzer device, calibrated using the sulfamethazine standard, was used to perform the elemental analysis. This analysis was performed by first weighing the sample in a tin capsule using a scale (Sartorius) to ensure it was less than 2 mg and then the capsule was placed in the automatic sampling system. The plant sample was then sent to the combustion reactor (1,050 °C) *via* the automatic sampling system and burned with $O_2$ accompanied by carrier gas. The % values of the elements were automatically calculated by the software device (*Kirsten, 1983*). Then, 3 g of each of the plant samples were dried, ground, weighed and then placed in a porcelain crucible and burned in a furnace set to 550 °C until white ash was obtained. After the combustion process completed, the sample was removed and weighed, and the difference between the initial weight and the final weight was considered the total ash ratio (*AOAC, 2000*). Crude oil analysis of the dried and ground plant samples was performed using the analytical methods reported by *AOAC (2000)*.

### Statistical analysis

The data obtained from the research were subjected to variance analysis using the SAS software (SAS Institute, Cary, NC, USA). Whether the difference between the obtained averages was significant or not was determined by the Duncan test. The mean squares and deterministic statistics of the data are given in Table 3.

## RESULTS

### Dry forage yield

Variance analysis results of dry forage yields are shown in Table 4. In both study years, dry forage yields increased with increased harvesting heights. According to the 2-year average
**Table 3 Deterministic statistics and mean square values of SSH and SS in 2020 and 2021 and 2-year average forage yields and quality.**

| Mean square | DHY | LC | LN | SC | SN | LCP | SCP | LCA | SCA | LCF | SCF |
|---|---|---|---|---|---|---|---|---|---|---|---|
| Year (Y) | 153,385.00 | 36.73 | 39.85 | 11.17 | 19.79 | 745.20 | 1,160.21 | 82.31 | 65.04 | 0.000734 | 0.103469 |
| Varieties (V) | 456,284.00 | 8.61 | 3.71 | 44.44 | 1.44 | 306.33 | 156.50 | 36.75 | 43.30 | 0.271096 | 2.481722 |
| Y * V | 905,792.00 | 14.98 | 2.59 | 12.45 | 2.70 | 396.02 | 204.05 | 116.91 | 12.06 | 0.095603 | 0.036475 |
| Harvesting height (HH) | 42,121,497.00 | 169.07 | 27.09 | 74.90 | 48.95 | 1,967.98 | 2,765.20 | 42.41 | 424.75 | 20.239297 | 13.550272 |
| Y * HH | 60,884.00 | 16.08 | 4.41 | 15.04 | 2.19 | 42.92 | 27.32 | 35.21 | 7.61 | 0.089009 | 0.039331 |
| V * HH | 7,964,635.00 | 65.94 | 2.76 | 58.83 | 6.60 | 154.39 | 203.06 | 58.17 | 51.72 | 0.321532 | 0.241861 |
| Y * V * HH | 687,806.00 | 32.15 | 5.64 | 16.25 | 6.10 | 70.69 | 193.81 | 43.90 | 94.76 | 0.631341 | 0.106625 |

| Mean ± SD | DHY | | C | | N | | CP | | CA | | CF |
|---|---|---|---|---|---|---|---|---|---|---|---|
| Deterministic statistics of variables for 2 years | | | | | | | | | | | |
| Stalk | 1,680.8 ± 608.5337 | | 38.807 ± 1.424 | | 1.248 ± 0.886 | | 6.917 ± 2.883 | | 8.341 ± 1.533 | | 1.732 ± 0.357 |
| Leaf | | | 41.243 ± 1.615 | | 2.332 ± 0.813 | | 11.782 ± 3.339 | | 11.007 ± 0.970 | | 2.759 ± 0.422 |

Note:
DHY, dry forage yield; LC, leaf carbon; SC, stalk carbon; LN, leaf nitrogen; SN, stalk nitrogen; LCP, leaf crude protein; SCP, stalk crude protein; LCA, leaf crude ash; SCA, stalk crude ash; LCF, leaf crude fat; SCF, stalk crude fat; C, carbon; N, nitrogen; CP, crude protein; CA, crude ash; CF, crude fat.

results, total dry hay yields increased with the increase in plant height at harvest. The lowest dry hay yield was obtained in the plots harvested from 30 cm height with 9,570.0 kg/da, while the highest value was determined in the plots harvested in the PMS with 26,050.0 kg/da. According to the varieties, the highest yields were determined in M81-E with 1,984.0 kg/da, Nutri Honey with 17,290.0 kg/da and Nutrima with 17,220.0 kg/da, while the lowest herbage yield was determined in Topper-76 variety with 15,880.0 kg/da. The total average dry forage yield was 16,480.0 kg/ha in the first year and increased to 17,140.0 kg/ha in the second year of the study (Table 4).

## Nutrient ingredients

### Leaf carbon ratio

Variance analysis results of the carbon contents of leaves are shown in Table 5. In general, the carbon content of leaves decreased with increased harvesting heights. The lowest leaf carbon ratios were measured in plants harvested at the PMS. The highest leaf carbon ratios were measured harvested at heights of 30, 60 and 90 cm. Leaf carbon ratios of the cultivars were between 40.92–41.60% of the 2 years. When combining cultivars with harvesting heights, Nutrima harvested at 120 cm in 2020 had a higher carbon content than other varieties in M81-E leaves harvested at 60 and 90 cm in 2021, and also the average of the 2 years. The average carbon content of the leaves in the first year was 41.75%, which decreased to 40.74% in the second year (Table 5).

### Stalk carbon ratio

Variance analysis results of the carbon contents of stalks are shown in Table 6. Carbon content in stalks increased with increased harvesting height. The highest carbon contents were observed in the thickest stalks and in stalks harvested at the PMS, as these stalks had the most mature cells. At the PMS, the average stalk carbon ratios were recorded average of 40.30% of the 2 years. The average stalk carbon ratios average range of 37.97–40.30% of the

**Table 4 Dry forage yields of SSH and SS varieties (kgha) in 2020 and 2021 and the 2-year averages.**

| Harvesting heights | Sorghum-Sudangrass (SSH) | | Sweet sorghum (SS) | | Average | 2020 year | 2021 |
|---|---|---|---|---|---|---|---|
| | Nutri honey | Nutrima | M81-E | Topper-76 | | | |
| **Two-years average (2020–2021)** | | | | | | | |
| 30 cm | 11,410.0[j] | 11,640.0[j] | 7,600.0[k] | 7,630.0[k] | 9,570.0[E] | **16,480.0[B]** | **17,140.0[A]** |
| 60 cm | 13,710.0[hı] | 14,580.0[gh] | 11,450.0[j] | 12,860.0[ij] | 13,150.0[D] | | |
| 90 cm | 13,480.0[hı] | 13,960.0[hı] | 14,160.0[hı] | 12,580.0[ij] | 13,550.0[D] | | |
| 120 cm | 21,750.0[c] | 18,190.0[de] | 17,560.0[ef] | 15,950.0[fg] | 18,360.0[C] | | |
| 150 cm | 21,340.0[c] | 22,940.0[j] | 17,240.0[ef] | 19,190.0[d] | 20,180.0[B] | | |
| PMS | 22,080.0[c] | 22,020.0[c] | 33,020.0[a] | 27,080.0[b] | 26,050.0[A] | | |
| Average | 17,290.0[A] | 17,220.0[A] | 19,840.0[A] | 15,880.0[B] | | | |
| Significant | PY: 0.0003, PV: 0.0001, PY * V: 0.0001, PHH: 0.0001, PY * HH: 0.3467, PV * HH: 0.0001, PY * V * HH: 0.0001 | | | | | | |

Note:
Y, years, V, varieties; HH, harvesting height; PMS, physiological maturity stage. In the table, lowercase letters indicate interactions and uppercase letters indicate differences between means. Bold letters and numbers indicate the significance between years.

**Table 5 Carbon ratios in the leaves of SSH and SS varieties (%) in 2020 and 2021 and the 2-year averages.**

| Harvesting heights | Sorghum-Sudangrass (SSH) | | Sweet sorghum (SS) | | Average | 2020 year | 2021 year |
|---|---|---|---|---|---|---|---|
| | Nutri honey | Nutrima | M81-E | Topper-76 | | | |
| **Two-years average (2020–2021)** | | | | | | | |
| 30 cm | 41.92[b-e] | 42.45[abc] | 40.31[g] | 41.97[a-e] | 41.66[AB] | **41.75[A]** | **40.74[B]** |
| 60 cm | 41.82[c-f] | 42.37[abc] | 43.18[a] | 41.58[c-f] | 42.23[A] | | |
| 90 cm | 41.63[c-f] | 41.57[c-f] | 43.07[ab] | 41.13[d-g] | 41.85[AB] | | |
| 120 cm | 41.08[efg] | 41.88[b-e] | 41.28[c-g] | 41.14[d-g] | 41.34[B] | | |
| 150 cm | 41.76[c-f] | 41.12[d-g] | 42.32[a-d] | 40.65[fg] | 41.46[B] | | |
| PMS | 39.01[h] | 40.23[g] | 37.36[ı] | 39.02[h] | 38.91[C] | | |
| Average | 41.20 | 41.60 | 41.25 | 40.92 | – | | |
| Significant | PY: 0.0001, PV: 0.0001, PY * V: 0.0001, PHH: 0.0001, PY * HH: 0.0001, PV * HH: 0.0001, PY * V * HH: 0.0001 | | | | | | |

Note:
Y, years; V, varieties; HH, harvesting height; PMS, physiological maturity stage. In the table, lowercase letters indicate interactions and uppercase letters indicate differences between means. Bold letters and numbers indicate the significance between years.

2 years. By variety, the lowest stalk carbon ratios were measured in Topper-76 (38.00%), with no significant difference in the stalk carbon ratios of all other cultivars. When combining cultivars with harvesting heights, the highest 2-year average stalk carbon ratio was measured in the M81-E variety harvested at the PMS, and the lowest carbon ratio was measured in the Topper-76 variety harvested at 90 cm, with no significant difference between cultivars harvested at 90, 120 and 150 cm heights. The carbon content of the stalks changed significantly between the 2 years of the study, with the average carbon ratio recorded at 38.53% in the first year and 39.09% in the second year of the study (Table 6).

*Leaf nitrogen ratio*

Variance analysis results of the nitrogen contents of leaves are shown in Table 7. The nitrogen content of leaves gradually decreased as the plants matured. The highest 2-year average nitrogen ratios were measured at 2.92% and 2.69% in the leaves of plants

**Table 6 Carbon ratios in the stalks of SSH and SS varieties (%) in 2020 and 2021 and the 2-year averages.**

| Harvesting heights | Sorghum-Sudangrass (SSH) | | Sweet sorghum (SS) | | Average | 2020 year | 2021 year |
|---|---|---|---|---|---|---|---|
| | Nutri honey | Nutrima | M81-E | Topper-76 | | | |
| Two-years average (2020–2021) | | | | | | | |
| 30 cm | 38.49$^{d\text{-}h}$ | 39.44$^{cde}$ | 37.61$^{hı}$ | 38.80$^{c\text{-}g}$ | 38.59$^B$ | **38.53$^B$** | **39.09$^A$** |
| 60 cm | 37.67$^{hı}$ | 38.71$^{c\text{-}h}$ | 39.37$^{cde}$ | 38.20$^{fgh}$ | 38.49$^{BC}$ | | |
| 90 cm | 38.06$^{f\text{-}ı}$ | 38.66$^{c\text{-}h}$ | 38.11$^{f\text{-}ı}$ | 37.03$^ı$ | 37.97$^C$ | | |
| 120 cm | 39.17$^{c\text{-}f}$ | 39.44$^{cde}$ | 38.20$^{fgh}$ | 37.64$^{hı}$ | 38.61$^B$ | | |
| 150 cm | 38.54$^{c\text{-}h}$ | 39.55$^{cd}$ | 39.61$^c$ | 37.91$^{ghı}$ | 38.90$^B$ | | |
| PMS | 39.66$^{bc}$ | 40.76$^b$ | 42.36$^a$ | 38.41$^{e\text{-}h}$ | 40.30$^A$ | | |
| Average | 38.60$^B$ | 39.43$^A$ | 39.21$^A$ | 38.00$^C$ | – | | |
| Significant | PY: 0.0001, PV: 0.0001, PY * V: 0.0004, PHH: 0.0001, PY * HH: 0.0005, PV * HH: 0.0001, PY * V * HH: 0.0537 | | | | | | |

Note:
Y, years; V, varieties; HH, harvesting height; PMS, physiological maturity stage. In the table, lowercase letters indicate interactions and uppercase letters indicate differences between means. Bold letters and numbers indicate the significance between years.

**Table 7 Nitrogen ratios in the leaves of SSH and SS varieties (%) in 2020 and 2021 and the 2-year averages.**

| Harvesting heights | Sorghum-Sudangrass (SSH) | | Sweet sorghum (SS) | | Average | 2020 year | 2021 year |
|---|---|---|---|---|---|---|---|
| | Nutri honey | Nutrima | M81-E | Topper-76 | | | |
| Two-years average (2020–2021) | | | | | | | |
| 30 cm | 3.02 | 2.88 | 2.59 | 3.18 | 2.92$^A$ | **2.87$^A$** | **1.81$^B$** |
| 60 cm | 2.91 | 2.82 | 2.40 | 2.63 | 2.69$^{AB}$ | | |
| 90 cm | 2.30 | 2.22 | 1.92 | 2.69 | 2.28$^{BC}$ | | |
| 120 cm | 1.90 | 2.33 | 1.98 | 2.44 | 2.16$^C$ | | |
| 150 cm | 2.63 | 2.31 | 2.31 | 2.33 | 2.40$^{BC}$ | | |
| PMS | 1.71 | 1.54 | 1.22 | 1.69 | 1.54$^D$ | | |
| Average | 2.41 | 2.35 | 2.07 | 2.49 | – | | |
| Significant | PY: 0.0001, PV: 0.0001, PY * V: 0.0004, PHH: 0.0001, PY * HH: 0.0005, PV * HH: 0.0001, PY * V * HH: 0.0537 | | | | | | |

Note:
Y, years; V, varieties; HH, harvesting height; PMS, physiological maturity stage. In the table, lowercase letters indicate interactions and uppercase letters indicate differences between means. Bold letters and numbers indicate the significance between years.

harvested at 30 and 60 cm, respectively, while the lowest nitrogen ratio was recorded as 1.54% in the leaves of plants harvested at their PMS. The nitrogen content of the leaves varied depending on the years. While the average nitrogen ratio was 2.87% in the first year, this value changed to 1.81% in the second year (Table 7).

### Stalk nitrogen ratio

Variance analysis results of stalk nitrogen content are shown in Table 8. In both study years, nitrogen content in the stalk decreased as the plants matured. The highest 2-year average nitrogen content was measured in the stalks of plants harvested at 30, 60 and 90 cm heights (1.86%, 1.65% and 1.72%, respectively). The lowest 2-year average stalk nitrogen ratio (0.23%) was measured in plants harvested at the PMS. The 2-year average stalk nitrogen ratio varied between 1.09–1.33% by cultivar. The average nitrogen content of stalks decreased significantly (by approximately 45%) in the second study year (Table 8).

**Table 8 Nitrogen ratios in the stalks of SSH and SS varieties (%) in 2020 and 2021 and the 2-year averages.**

| Harvesting heights | Sorghum-Sudangrass (SSH) | | Sweet sorghum (SS) | | Average | 2020 year | 2021 year |
|---|---|---|---|---|---|---|---|
| | Nutri honey | Nutrima | M81-E | Topper-76 | | | |
| **Two-years average (2020–2021)** | | | | | | | |
| 30 cm | 1.68 | 1.95 | 2.00 | 1.80 | 1.86[A] | **1.62[A]** | **0.88[B]** |
| 60 cm | 1.87 | 1.81 | 1.14 | 1.78 | 1.65[AB] | | |
| 90 cm | 1.97 | 1.43 | 1.66 | 1.81 | 1.72[A] | | |
| 120 cm | 1.76 | 1.44 | 0.81 | 1.15 | 1.29[B] | | |
| 150 cm | 0.58 | 0.65 | 0.90 | 0.85 | 0.75[C] | | |
| PMS | 0.14 | 0.18 | 0.02 | 0.60 | 0.23[D] | | |
| Average | 1.33 | 1.24 | 1.09 | 1.33 | – | | |
| Significant | PY: 0.0001, PV: 0.1254, PY * V: 0.0153, PHH: 0.0001, PY * HH: 0.1240, PV * HH: 0.0479, PY * V * HH: 0.0748 | | | | | | |

Note:
   Y, years; V, varieties; HH, harvesting height; PMS, physiological maturity stage. In the table, lowercase letters indicate interactions and uppercase letters indicate differences between means. Bold letters and numbers indicate the significance between years.

### Leaf crude protein ratio

Variance analysis results of the crude protein contents of leaves are shown in Table 9. The highest 2-year average crude protein content was found in Nutri Honey and Topper-76 with values of 12.49% and 11.98%, respectively, followed by the Nutrima and M81-E varieties at 10.67% and 10.55%, respectively. By harvesting height, the highest 2-year average crude protein content in leaves was 14.27% in plants harvested at 30 cm, while the lowest was recorded as 6.46% in plots harvested at their PMS. The average crude protein content of leaves decreased 20% between 2020 and 2021 (Table 9).

### Stalk crude protein ratio

Variance analysis results of the crude protein contents of stalks are shown in Table 10. The highest 2-year average crude protein ratio in stalks by variety was 7.43% in the Topper-76 variety, while the lowest was 5.88% in the Nutri Honey variety. The average stalk crude protein content decreased with harvest height. The highest crude protein content was 10.33% in plants harvested at 30 cm height, while the lowest values were 3.57% and 4.60% in plants harvested at PMS and 150 cm height. The average crude protein content of the stalks decreased up to 20% in the second study year (Table 10).

### Leaf crude ash ratio

Variance analysis results of the crude ash contents of leaves are given in Table 11. By variety, the highest 2-year average ratio of crude ash in the leaves was 11.77% in the Nutri Honey variety, followed by Topper-76 with 11.28%, M81-E with 11.11% and Nutrima with 11.02%. By harvesting height, the crude ash content of the leaves was 11.36% in plants harvested at 30 cm and 10.94%, 10.96% and 12.05% in plants harvested at 90, 120 cm and PMS, respectively, and there was significant difference. The average crude ash content of leaves was 10.86% in the first year, which increased to 11.73% in the second study year (Table 11).

**Table 9 Crude protein ratios in the leaves of SSH and SS varieties (%) in 2020 and 2021 and the 2-year averages.**

| Harvesting heights | Sorghum-Sudangrass (SSH) | | Sweet sorghum (SS) | | Average | 2020 year | 2021 year |
|---|---|---|---|---|---|---|---|
| | Nutri honey | Nutrima | M81-E | Topper-76 | | | |
| **Two-years average (2020–2021)** | | | | | | | |
| 30 cm | 14.49 | 14.20 | 13.28 | 15.10 | 14.27$^A$ | **12.72$^A$** | **10.12$^B$** |
| 60 cm | 13.93 | 12.65 | 12.20 | 13.81 | 13.15$^B$ | | |
| 90 cm | 13.26 | 11.28 | 10.89 | 12.76 | 12.05$^C$ | | |
| 120 cm | 12.83 | 10.19 | 11.35 | 12.72 | 11.77$^{CD}$ | | |
| 150 cm | 11.64 | 8.91 | 10.63 | 12.08 | 10.82$^D$ | | |
| PMS | 8.76 | 6.76 | 4.92 | 5.41 | 6.46$^E$ | | |
| Average | 12.49$^A$ | 10.67$^B$ | 10.55$^B$ | 11.98$^A$ | – | | |
| Significant | PY: 0.0001, PV: 0.0001, PY * V: 0.0001, PHH: 0.0001, PY * HH: 0.3118, PV * HH: 0.1284, PY * V * HH: 0.8293 | | | | | | |

Note:
Y, years; V, varieties; HH, harvesting height; PMS, physiological maturity stage. In the table, lowercase letters indicate interactions and uppercase letters indicate differences between means. Bold letters and numbers indicate the significance between years.

**Table 10 Crude protein ratios in the stalks of SSH and SS varieties (%) in 2020 and 2021 and the 2-year averages.**

| Harvesting heights | Sorghum-Sudangrass (SSH) | | Sweet sorghum (SS) | | Average | 2020 year | 2021 year |
|---|---|---|---|---|---|---|---|
| | Nutri honey | Nutrima | M81-E | Topper-76 | | | |
| **Two-years average (2020–2021)** | | | | | | | |
| 30 cm | 10.10 | 11.63 | 8.96 | 10.67 | 10.33$^A$ | **8.38$^A$** | **5.13$^B$** |
| 60 cm | 7.88 | 9.54 | 8.17 | 9.54 | 8.78$^B$ | | |
| 90 cm | 6.58 | 8.48 | 6.32 | 8.18 | 7.39$^C$ | | |
| 120 cm | 5.16 | 6.23 | 6.10 | 5.86 | 5.84$^D$ | | |
| 150 cm | 3.25 | 3.72 | 5.33 | 6.07 | 4.60$^E$ | | |
| PMS | 2.30 | 3.30 | 4.43 | 4.27 | 3.57$^E$ | | |
| Average | 5.88$^C$ | 7.15$^{AB}$ | 6.55$^{BC}$ | 7.43$^A$ | – | | |
| Significant | PY: 0.0001, PV: 0.0001, PY * V: 0.0001, PHH: 0.0001, PY * HH: 0.4951, PV * HH: 0.0064, PY * V * HH: 0.0100 | | | | | | |

Note:
Y, years; V, varieties; HH, harvesting height; PMS, physiological maturity stage. In the table, lowercase letters indicate interactions and uppercase letters indicate differences between means. Bold letters and numbers indicate the significance between years.

### Stalk crude ash ratio

Variance analysis results of the crude ash contents of stalks are given in Table 12. The crude ash content of the stalk decreased with the increase in plant height at harvest. The highest crude ash content (9.82%) was determined in the plots harvest at 30 cm height and the lowest (6.49%) was determined in the harvests made during PMS period. According to the varieties, the highest crude ash content was determined in Nutrima and Nutri Honey varieties with 8.90% and 8.61%, respectively, and the lowest was determined in Topper-76 variety with 8.04%. There were significant changes in the crude ash content of the stalk according to the years. Accordingly, while this ratio was 8.88% in the first year, it decreased to 8.11% in the second year (Table 12).

**Table 11 Crude ash ratios in the leaves of SSH and SS varieties (%) in 2020 and 2021 and the 2-year averages.**

| Harvesting heights | Sorghum-Sudangrass (SSH) | | Sweet sorghum (SS) | | Average | 2020 year | 2021 year |
|---|---|---|---|---|---|---|---|
| | Nutri honey | Nutrima | M81-E | Topper-76 | | | |
| **Two-years average (2020–2021)** | | | | | | | |
| 30 cm | 11.58$^{cde}$ | 11.43$^{c-g}$ | 11.35$^{c-h}$ | 11.08$^{e-1}$ | 11.36$^{B}$ | **10.86$^{B}$** | **11.73$^{A}$** |
| 60 cm | 11.77$^{bcd}$ | 11.06$^{e-1}$ | 10.58$^{ij}$ | 10.94$^{f-1}$ | 11.09$^{BCD}$ | | |
| 90 cm | 11.67$^{b-e}$ | 11.20$^{d-1}$ | 10.24$^{j}$ | 10.66$^{ij}$ | 10.94$^{D}$ | | |
| 120 cm | 11.57$^{b-f}$ | 10.66$^{ij}$ | 10.88$^{f-j}$ | 10.74$^{g-1}$ | 10.96$^{CD}$ | | |
| 150 cm | 12.02$^{bc}$ | 11.25$^{c-1}$ | 11.03$^{d-j}$ | 11.13$^{hij}$ | 11.36$^{BC}$ | | |
| PMS | 12.00$^{a-e}$ | 10.53$^{gij}$ | 12.57$^{ab}$ | 13.12$^{a}$ | 12.05$^{A}$ | | |
| Average | 11.77$^{A}$ | 11.02$^{B}$ | 11.11$^{B}$ | 11.28$^{B}$ | – | | |
| Significant | PY: 0.0001, PV: 0.0001, PY * V: 0.0001, PHH: 0.0001, PY * HH: 0.0001, PV * HH: 0.0001, PY * V * HH: 0.0041 | | | | | | |

Note:
Y, years; V, varieties; HH, harvesting height; PMS, physiological maturity stage. In the table, lowercase letters indicate interactions and uppercase letters indicate differences between means. Bold letters and numbers indicate the significance between years.

**Table 12 Crude ash ratios in the stalks of SSH and SS varieties (%) in 2020 and 2021 and the 2-year averages.**

| Harvesting heights | Sorghum-Sudangrass (SSH) | | Sweet sorghum (SS) | | Average | 2020 year | 2021 year |
|---|---|---|---|---|---|---|---|
| | Nutri honey | Nutrima | M81-E | Topper-76 | | | |
| **Two-years average (2020–2021)** | | | | | | | |
| 30 cm | 9.98$^{ab}$ | 10.33$^{a}$ | 9.58$^{bcd}$ | 9.40$^{cde}$ | 9.82$^{A}$ | **8.88$^{A}$** | **8.11$^{B}$** |
| 60 cm | 9.58$^{bcd}$ | 9.92$^{abc}$ | 9.02$^{def}$ | 9.05$^{def}$ | 9.39$^{B}$ | | |
| 90 cm | 9.25$^{c-f}$ | 9.51$^{b-e}$ | 8.11$^{h-k}$ | 8.24$^{gh1}$ | 8.78$^{C}$ | | |
| 120 cm | 8.87$^{efg}$ | 8.71$^{fgh}$ | 8.95$^{d-g}$ | 7.36$^{jkl}$ | 8.47$^{C}$ | | |
| 150 cm | 7.96$^{h-k}$ | 7.82$^{ijk}$ | 8.17$^{g-k}$ | 7.98$^{h-k}$ | 7.98$^{D}$ | | |
| PMS | 6.00$^{m}$ | 7.14$^{j-m}$ | 6.64$^{lm}$ | 6.21$^{m}$ | 6.49$^{E}$ | | |
| Average | 8.61$^{AB}$ | 8.90$^{A}$ | 8.41$^{B}$ | 8.04$^{C}$ | – | | |
| Significant | PY: 0.0001, PV: 0.0001, PY * V: 0.0315, PHH: 0.0001, PY * HH: 0.3462, PV * HH: 0.0011, PY * V * HH: 0.0001 | | | | | | |

Note:
Y, years; V, varieties; HH, harvesting height; PMS, physiological maturity stage. In the table, lowercase letters indicate interactions and uppercase letters indicate differences between means. Bold letters and numbers indicate the significance between years.

### Leaves crude fat ratio

Variance analysis results of the crude fat contents of leaves are given in Table 13. The highest 2-year crude fat ratio (3.39%) was found in the leaves of plants harvested at the lowest height (30 cm), while the lowest value (2.24%) was obtained from the leaves of plants harvested at the highest height (PMS). In the second study year, the average crude fat ratios of leaves varied between 2.71–2.82% by variety and between 2.22–3.49% when combining variety with harvesting heights. There were no significant changes in the crude fat content of the leaves according to the years (Table 13).

### Stalk crude fat ratio

Variance analysis results of the crude fat contents of the stalks are given in Table 14. As with the crude fat content of leaves, the crude fat content of the stalks decreased as the plants matured. According to the varieties, the highest crude fat content of the stalk was

**Table 13 Crude fat ratios in the leaves of SSH and SS varieties (%) in 2020 and 2021 and the 2-year averages.**

| Harvesting heights | Sorghum-Sudangrass (SSH) | | Sweet sorghum (SS) | | Average | 2020 year | 2021 year |
|---|---|---|---|---|---|---|---|
| | Nutri honey | Nutrima | M81-E | Topper-76 | | | |
| **Two-years average (2020–2021)** | | | | | | | |
| 30 cm | 3.36 | 3.33 | 3.49 | 3.38 | 3.39$^A$ | **2.76** | **2.76** |
| 60 cm | 3.16 | 2.92 | 3.08 | 2.96 | 3.03$^B$ | | |
| 90 cm | 2.86 | 2.82 | 2.80 | 2.70 | 2.80$^C$ | | |
| 120 cm | 2.75 | 2.56 | 2.60 | 2.60 | 2.63$^D$ | | |
| 150 cm | 2.51 | 2.51 | 2.46 | 2.39 | 2.47$^E$ | | |
| PMS | 2.24 | 2.22 | 2.28 | 2.22 | 2.24$^F$ | | |
| Average | 2.82 | 2.73 | 2.79 | 2.71 | – | | |
| Significant | PY: 0.8966, PV: 0.1068, PY * V: 0.5319, PHH: 0.0001, PY * HH: 0.8389, PV * HH: 0.9364, PY * V * HH: 0.4876 | | | | | | |

Note:
Y, years; V, varieties; HH, harvesting height; PMS, physiological maturity stage. In the table, lowercase letters indicate interactions and uppercase letters indicate differences between means. Bold letters and numbers indicate the significance between years.

**Table 14 Crude fat ratios in the stalks of SSH and SS varieties (%) in 2020 and 2021 and the 2-year averages.**

| Harvesting heights | Sorghum-Sudangrass (SSH) | | Sweet sorghum (SS) | | Average | 2020 year | 2021 year |
|---|---|---|---|---|---|---|---|
| | Nutri honey | Nutrima | M81-E | Topper-76 | | | |
| **Two-years average (2020–2021)** | | | | | | | |
| 30 cm | 2.14 | 1.98 | 2.44 | 2.29 | 2.21$^A$ | **1.76$^A$** | **1.71$^B$** |
| 60 cm | 1.92 | 1.81 | 2.23 | 2.04 | 2.00$^B$ | | |
| 90 cm | 1.74 | 1.61 | 2.02 | 1.74 | 1.77$^C$ | | |
| 120 cm | 1.55 | 1.53 | 1.84 | 1.58 | 1.62$^D$ | | |
| 150 cm | 1.45 | 1.33 | 1.69 | 1.48 | 1.49$^E$ | | |
| PMS | 1.25 | 1.22 | 1.41 | 1.31 | 1.30$^F$ | | |
| Average | 1.67$^C$ | 1.58$^D$ | 1.94$^A$ | 1.74$^B$ | – | | |
| Significant | PY: 0.0223, PV: 0.0001, PY * V: 0.5999, PHH: 0.0001, PY * HH: 0.8440, PV * HH: 0.6417, PY * V * HH: 0.9842 | | | | | | |

Note:
Y, years; V, varieties; HH, harvesting height; PMS, physiological maturity stage. In the table, lowercase letters indicate interactions and uppercase letters indicate differences between means. Bold letters and numbers indicate the significance between years.

determined in M-81E variety with 1.94% and the lowest was determined in Nutri Honey variety with 1.67%. There was a decrease in the crude fat content of the stalk due to the increase in plant height at harvest. The highest crude fat content was determined in plots harvest at 30 cm height with 2.21%, while the lowest value was determined in plots harvest at 1.30% PMS period. There were significant changes in the crude fat content of the stalk according to the years. While the crude fat content was 1.76% in the first year, this ratio decreased to 1.71% in the second year (Table 14).

## DISCUSSION

Most crops follow a similar growth pattern, beginning with slow growth followed by rapid growth before slowing down again towards the end of their growth cycle (*Altın, Gökkuş & Koç, 2011*). Since there are fewer chloroplasts at the beginning of the growth cycle there are

also fewer assimilation products. But as growth progresses, the number of chloroplasts increases, which increases photosynthetic processes, producing more organic mass. Vegetative growth continuously increases until the generative period. During the generative period, the products of photosynthesis are transferred to generative organs instead of vegetative tissues such as the roots, shoots or leaves (*Altın, Gökkuş & Koç, 2011*). This limits the forage yield at this stage of development (*Chattha et al., 2017*). Because of this, the dry forage yields increased regularly and continuously between harvesting heights of 30 cm to the PMS of the crop in this experiment (Table 4). Previous studies have found differing yield values in experiments carried out with sorghum in Türkiye. For example, *Aydınoğlu & Çakmakçı (2018)* found an average fresh forage yield between 46,000.0–81,880.0 kg/ha and dry forage yield between 11,870.0–20,370.0 kg/ha. A separate study conducted with 13 different sorghum cultivars in Bingöl found an average fresh forage yield of 73,230.0 kg/ha and an average dry forage yield of 13,080.0 kg/ha (*Özmen, 2017*). Other studies conducted in different regions found average fresh forage yields between 146,410.0–190,300.0 kg/ha in Konya (*Acar, Akbudak & Sade, 2002*), 22,890.0–47,170.0 kg/ha in Çanakkale (*Küçüksemerci & Baytekin, 2017*), 67,300.0 kg/ha (*Sürmen & Kara, 2022*) and 46,500.0–62,600.0 kg/ha in Aydın (*Çelik & Türk, 2021*). Studies have also found average dry forage yields varying between 13,080.0 kg/ha (*Özmen, 2017*), 61,00.0–11,830.0 kg/ha (*Tosunoğlu, 2014*), 16,540.0 kg/ha (*Çeçen, Öten & Erdurmuş, 2005*), 8,100.0–21,100.0 kg/ha (*Kara, Sürmen & Erdoğan, 2019*), 13,500.0–28,400.0 kg/ha (*Kır & Şahan, 2019*) and 14,800 kg/ha (*Sürmen & Kara, 2022*). Forage yields differ because different varieties have different genetic structures and react differently to environmental factors (*Özyazıcı & Açıkbaş, 2019*). Differences in forage yield by variety was also observed in this study, with the SS Topper-76 variety yielding significantly less than other varieties (Table 4).

The carbon and nitrogen contents of plants vary by species and varieties, and even between different organs of the same species (*Yao et al., 2015*; *Suhui et al., 2018*; *Zhang et al., 2018*). The nitrogen content of plants is closely related to photosynthesis and plant respiration meaning plants have high nitrogen content at the beginning of the growth cycle because there is more photosynthesis and respiration activity in the leaves, and nitrogen and carbon content decreases as the growth cycle continues (*Zhang et al., 2018*, *2020*). In this study, the carbon ratios of leaves and stalks of different sorghum cultivars differed depending on plant height, with the carbon ratio of the leaves decreasing as plant height increased and the carbon ratio of the stalks increasing as plant height increased. The carbon ratio in the leaves decreased by approximately of 6% from the beginning to the end of the growth cycle, while the carbon ratio of the stalks increased by 4% during the growth cycle. Conversely, the nitrogen ratios in both the leaves and stalks decreased-by 47% in leaves and by 87% in stalks-from the beginning to the end of the growth cycle (Tables 5–8). The nitrogen content of a plant is closely related to its carbon ratio. High nitrogen concentration in the plant results in higher photosynthetic capacity, which leads to higher respiration (*Chapin et al., 1987*; *Lambers et al., 1989*). Accordingly, more nitrogen is used in photosynthesis and more carbon in respiration (*Poorter & Remkes, 1990*). The photosynthetic capacity of plants is higher in earlier growth stages compared to

late stages because the number of leaves is higher in the earlier stages, and leaves have high photosynthetic capacity and more nitrogen and nutrient content than the stalks and roots (*Poorter, Remkes & Lambers, 1990*). The position of the leaves, the age of the plant and certain environmental factors affect a plant's photosynthetic capacity (*Aighewi & Ekanayake, 2004*; *Hgaza et al., 2009*).

In this study, the crude protein ratios of sorghum cultivars showed significant variation between the leaves and the stalks. The 2-year average crude protein content was 11.42% in leaves and 6.75% in the stalks, meaning the crude protein content in leaves was approximately 40% higher than in the stalks. By variety, the highest crude protein content was found in the leaves of the Nutri Honey and Topper-76 varieties and the stalks of the Topper-76 variety. Crude protein contents of leaves and stalks proportionally decreased, by a total of 64% in leaves and 60% in the stalks of sorghum varieties, with increased plant height (Tables 9 and 10). Many factors affect forage quality in forage crops, with nutritional value considered the most important factor. The nutritional value of hay is measured by crude protein content. If the crude protein content of forage plants is 12% or lower, the quality of the hay is considered low. A crude protein content of 15% is considered medium quality hay, and hay with a crude protein content 18% and above is considered to have high nutritional quality (*Budak & Budak, 2014*; *Cherney & Hall, 2014*). Plants contain more crude protein in their leaves because of the higher photosynthesis capacity and the higher nitrogen and nutrient content of leaves compared to the stalks and roots (*Poorter, Remkes & Lambers, 1990*). Previous studies have shown that the crude protein ratios of leaves and stalks decrease with an increase in harvesting heights (*Keskin, Yılmaz & Akdeniz, 2005*; *Karataş & Tansı, 2011*). The results of this study align with previous studies; the highest crude protein content was observed at the beginning of the growth cycle when sorghum crops were young. In general, plants possess more dividing cells at the beginning of their growth (*Taiz & Zeiger, 2008*), and these cells have higher levels of physiological activity. All of the biochemical reactions in living things take place under the catalysis of enzymes. Enzymes are made up of proteins, so the protein ratio is always high at the beginning of the growth cycle in plants (*Gökkuş et al., 2011*). Crude protein ratio also decreases due to the decrease in physiological activity that occurs with plant growth (*Towne & Ohlenbusch, 1992*). In this study, the average crude protein contents ranged from 10.67–12.49% in leaves and 5.88–7.43% in stems. A separate study recorded the crude protein content of leaves between 14–15% and between 3–4% in stems (*Karataş & Tansı, 2011*). One previous study on four different sorghum varieties determined that the average crude protein ratio of the whole plant is around 5% (*Keskin, Yılmaz & Akdeniz, 2005*), while another study conducted on 13 different sorghum cultivars in the Bingöl province of Turkiye recorded the average crude protein content of sorghum cultivars as 4.81% (*Özmen, 2017*). The average crude protein ratio has been measured in previous studies as 9.5–10.2% (*Kozłowski et al., 2006*), 7.2% (*Marsalis et al., 2010*), 7.1–9.7% (*Arslan & Çakmakçı, 2011*), 7.2–8.8% (*Canbolat, 2012*) and 8.3% (*Tosunoğlu, 2014*).

Crude ash (mineral element) contents of different sorghum cultivars showed significant variation in plant leaves and stalks, and by variety and harvesting height. Generally, the
macro and micro element contents of the leaves were found to be higher than that of the stalks. Previous studies have determined that the protein, vitamin and mineral contents of leaves are higher than that of the stalks, while cellulose, hemicellulose and lignin contents are lower (*Fales & Fritz, 2007*; *Jung, 2012*; *Temel & Keskin, 2020*). This explains the high crude ash content of the leaves observed in this study. Crude ash contents of sorghum cultivars also differed significantly from each other, likely because of genetic variations in different varieties of sorghum cultivars (*Khan et al., 2006*; *Kering et al., 2011*; *Özyazıcı & Açıkbaş, 2019*). The SSH cultivars generally contained higher crude ash than that of the sudangrass cultivar (Tables 11 and 12). This is likely related to the amount of nutrients taken from the soil along with the genetic differences of these varieties (*Özyazıcı & Açıkbaş, 2020*). Previous studies have found that the ratios of P, K, Ca and Mg significantly differ between sorghum cultivars (*Başbağ et al., 2011*; *Gülümser et al., 2017*; *Gürsoy & Macit, 2017*; *Başbağ, Çaçan & Sayar, 2018*; *Polat & Bayraklı, 2019*), and even between different varieties within the same cultivar (*Lema, Cebert & Sapra, 2004*; *Markovic et al., 2014*; *Engin & Mut, 2018*; *Özyazıcı et al., 2018a*, *2018b*; *Turan et al., 2018*; *Özyazıcı & Açıkbaş, 2019*; *Tan, 2019*; *Özyazıcı & Açıkbaş, 2021*). In this study, the crude ash content of SSH and SS cultivars decreased with an increase in crop height, meaning mineral element contents decreased based on the physiological maturity of crops. Crops need high levels of crude ash, especially in times of rapid growth (Tables 11 and 12). Because most of the crude ash is found in the protoplasm, where physiological activities are intense, and less in the cell wall (*Spears, 1994*), the crude ash content decreases with growth, but the ratio of total organic matter to mineral matter increases because cell wall compounds increase during plant growth. Another reason for the decrease in crude ash content may be the increased amount of dry matter that occurs during plant growth (*Kaçar, 2012*). Some studies (*Dactylis glomerata* L., *Lathyrus sativus* L.) have reported that decreases in macro and micro element contents are correlated with the ripening process in crops (*Schlegel et al., 2016*; *Can & Ayan, 2017*; *Özyazıcı & Açıkbaş, 2020*). According to a study conducted in rangelands, crude ash content decreases because of the physiological maturity of plants (*Gökkuş et al., 2011*).

In this study, crude fat ratios were significantly different in leaves and stalks, and by cultivar and harvesting heights. Only the fat content of the stalks differed significantly by cultivar, with the stalks of SS cultivars having 11.4% higher fat content than that of the SSH cultivars (Tables 13 and 14). This difference is likely due to the genetic differences of the cultivars (*Özyazıcı & Açıkbaş, 2020*). In addition to this, the crude fat contents of the stalks and leaves of the crops also differed significantly. Generally, the average crude fat content of the leaves was 37.3% higher than that of the stalks because the metabolic activities in the leaves are faster and more abundant than those in the stalks (Tables 13 and 14). A high leaf ratio has been reported to be important to forage quality in forage crops because there are close relationships between leaf ratios and crude protein, digestible crude protein, ratios of mineral substances and digestibility of dry matter (*Açıkgöz, 2001*). In this study, the crude fat levels of leaves and stalks decreased by approximately 33.9% and 41.2%, respectively, as the plant height increased. Fats are important to the functions and continuity of the biological membranes in crops, the formation of nutrients stored in the

seeds, and the stability of the protoplasm. Fats found in the roots, stalks, leaves, flowers and seeds of the crops protect these organs against external factors. The ratio of crude fat can vary by plant organ, variety, species, genotype, and developmental phase/stage (*Pallardy, 2008*). In similar studies, the average crude fat content of 58 sorghum genotypes and three sorghum cultivars varied between 2.32–5.74% (*Kaplan et al., 2016*). Other studies have reported average crude fat content of 5.63% in one sorghum variety (*Osman et al., 2000*), between 1.5–2.23% in three sorghum varieties (*Pontieri et al., 2021*) and between 2.6–3.1% in sorghum varieties (*Canbolat, 2012*).

## CONCLUSIONS

This study aimed to determine the variation in summer main crop forage production, growth process, carbon and nitrogen content and forage quality characteristics of different harvesting heights, time and different varieties of sweet sorghum (SS) and sorghum sudangrass hybrid (SSH) cultivars. The obtained results of this research were as follows. There has been an increase in hay yields depending on the increase in plant height and time. The highest yields were determined in M81-E, Nutrima and Nutri Honey varieties. Nutrient contents (carbon, nitrogen, crude protein, crude ash and crude fat) of leaves and stalk decreased with the increase in plant height at harvesting period, while only carbon contents of stalk and crude ash contents of leaves increased. Hay yields increased with the increase in plant height at harvesting time, while nutrient contents of hay decreased. It is recommended to cultivated Nutri Honey and M81-E sorghum varieties taking into account the hay yield and the nutritional values of the varieties, and harvest time them when they reach 90 and 120 cm height.

### Funding

This research was funded by the "Scientific and Technological Research Council of Turkey (TÜ-BİTAK)", Grant Number: 120 O 527. The funders had no role in study design, data collection and analysis, decision to publish, or preparation of the manuscript.

### Grant Disclosures

The following grant information was disclosed by the authors:
Scientific and Technological Research Council of Turkey (TÜ-BİTAK): 120 O 527.

### Competing Interests

The authors declare that they have no competing interests.

### Author Contributions

- Fırat Alatürk conceived and designed the experiments, performed the experiments, analyzed the data, prepared figures and/or tables, authored or reviewed drafts of the article, and approved the final draft.

## Data Availability

The raw data are available in the Supplemental Files.

## Supplemental Information

Supplemental information for this article can be found online at http://dx.doi.org/10.7717/peerj.17274#supplemental-information.

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
