# Peer review of "Effects of harvest height and time on hay yield and quality of some sweet sorghum and sorghum Sudangrass hybrid varieties"

_PeerJ, doi:10.7717/peerj.17274_

## Round 0.1 · original submission · Major Revisions

The authors need to include suggestions given by reviewers.

**Language Note:** The review process has identified that the English language must be improved. PeerJ can provide language editing services - please contact us at [email protected] for pricing (be sure to provide your manuscript number and title). Alternatively, you should make your own arrangements to improve the language quality and provide details in your response letter. – PeerJ Staff

·

Basic reporting

The experiment investigated the effect/influence of different harvesting time points on the yield and nutritional composition of sweet sorghum and sorghum Sudangrass hybrid varieties. The study interests the scientific community since it generates important information on sorghum as a forage crop. However, the author needs to pay attention to some of my comments to improve the quality of the manuscript for publication

General comment(s)
1. The English language should be improved throughout the entire manuscript to ensure readers understand the written texts clearly. I suggest the author use free online tools like Grammarly or seek assistance from a fluent English speaker familiar with scientific writing.
2. The title of the manuscript needs to be changed for clarity. For instance:
Influence of different cutting time-points on the yield and forage quality of sweet sorghum and sorghum Sudangrass hybrid varieties

Introduction
Sorghum x Sudan grass hybrids are indeed very important forage crops used in animal nutrition. However, the success of a forage crop in animal nutrition depends on several factors among which is lignin (Kir et al. 2019; Kaplan et al. 2019). The author needs to write a short paragraph on the influence of lignin content on forage quality of sorghum

Experimental design

Materials and methods
L77 – 81: The author needs to briefly describe how soil samples were obtained. For instance, were soil samples collected randomly from the field and pooled for analysis? How was the elemental composition (organic matter, phosphorous, and potassium determined?

L100 – 103: The experimental layout has been well described. However, it would be better if the author could illustrate with a schematic diagram of the layout. This will reduce the number of words written to describe the design.

L114: How many plants per variety were cut to obtain the yield? Although dry yields were reported, why didn’t the author report data on fresh yields? Fresh yield data is also important to help the reader know how much water each variety had and this correlates with the concentration of available nutrients within the forage crop.

Validity of the findings

Results
Figure 1: Please provide a better quality figure (300 dpi). The title of the figure should change to something that is clearer, such as; average temperature and precipitation values recorded during the experimental period.
Tables: The information in the tables has been presented well, however, there is a need to report results as means ± standard deviation (SD) or as means ± standard error of mean (SEM). This will help the reader to have a full picture of how the mean values deviated from the true mean or how they are close to the true mean. Likewise, the author should include legends on all the tables. Readers need to understand what the average values (means) with different or the same lower superscript letters mean. For instance, do average values (means) with different lower superscript letters significantly different?

Discussion
Generally, there is a need to improve the sentence structure and grammar throughout the discussion.
L382 – 388: What is the minimum crude protein content to indicate a good quality forage crop? More information is needed.
L435: The author reports on the mineral content of the tested varieties but the reader lacks an idea of which exact minerals were being discussed. The author should specify which mineral elements were being discussed and also present their results within the results section of the manuscript.
L444 – 449: The author should report the crops (scientific names) that were reported in other studies

Conclusion
The entire conclusion needs to be shortened (no need to report on the experimental design and aim of the study) and restructured to summarize the main results of the study. For instance, the author should report the varieties/variety with the highest dry yield, crude protein content of leaves and stalks, crude fat, C and N in plant tissues, and a variety or varieties that showed the lowest values. Likewise, the author should indicate the best plant growth stage at which the best forage qualities (crude protein, crude fat, etc.) were observed and any recommendations if any.

Additional comments

No comment

·

Basic reporting

The results section is too lengthy and leads to an imbalance in the proportion of the entire article. The results should be streamlined and the discussion should be expanded appropriately. The literature cited in the article is too old. Try to cite the latest research articles or those in recent years.

Experimental design

Why was two years of data separated for statistical processing? It is desirable to perform statistical processing only on data presented as averages.

Validity of the findings

Although it is a material used as forage, there is no analysis of the forage quality associated with forage. Analysis of the fiber composition (ADF and NDF, etc.) must be presented.

Additional comments

None

---

## Round 0.2 · Major Revisions

The authors need to check all my comments given in the attached manuscript PDF.

·

Basic reporting

the manuscript has been revised according to my previous comments

Experimental design

the manuscript has been revised according to my previous comments

Validity of the findings

the manuscript has been revised according to my previous comments

Additional comments

the manuscript has been revised according to my previous comments

---

## Round 0.3 · Minor Revisions

The authors need to include suggestions given by reviewer 2.

·

Basic reporting

No comment

Experimental design

No comment

Validity of the findings

The selection of varieties suitable for the region and the cutting height were appropriately determined.

Additional comments

See attached PDF

---

## Round 0.4 · accepted · Accept

The revised manuscript seems good. Check all references, figures and tables carefully.